# Recent Development in the Understanding of Molecular and Cellular Mechanisms Underlying the Etiopathogenesis of Alzheimer’s Disease

**DOI:** 10.3390/ijms24087258

**Published:** 2023-04-14

**Authors:** Atefeh Afsar, Maria del Carmen Chacon Castro, Adedamola Saidi Soladogun, Li Zhang

**Affiliations:** Department of Biological Sciences, The University of Texas at Dallas, Richardson, TX 75080, USA

**Keywords:** Alzheimer’s disease, amyloid β-protein, tau proteins, heme, heme oxygenase

## Abstract

Alzheimer’s disease (AD) is a progressive neurodegenerative disorder that leads to dementia and patient death. AD is characterized by intracellular neurofibrillary tangles, extracellular amyloid beta (Aβ) plaque deposition, and neurodegeneration. Diverse alterations have been associated with AD progression, including genetic mutations, neuroinflammation, blood–brain barrier (BBB) impairment, mitochondrial dysfunction, oxidative stress, and metal ion imbalance.Additionally, recent studies have shown an association between altered heme metabolism and AD. Unfortunately, decades of research and drug development have not produced any effective treatments for AD. Therefore, understanding the cellular and molecular mechanisms underlying AD pathology and identifying potential therapeutic targets are crucial for AD drug development. This review discusses the most common alterations associated with AD and promising therapeutic targets for AD drug discovery. Furthermore, it highlights the role of heme in AD development and summarizes mathematical models of AD, including a stochastic mathematical model of AD and mathematical models of the effect of Aβ on AD. We also summarize the potential treatment strategies that these models can offer in clinical trials.

## 1. Introduction

In older people, Alzheimer’s disease (AD) is now the most frequent type of dementia. Unfortunately, existing treatments are not highly effective. Therefore, it is crucial to increase our understanding of the mechanisms behind the emergence and progression of AD [1]. With 35.6 million cases globally, AD is considered one of the most common forms of aging-related dementia, and the number of cases is expected to triple or quadruple by 2050. AD impairs memory, cognition, and behavior, and ultimately results in death [2]. The accumulation of the extracellular Aβ protein both inside and outside neurons and the existence of hyperphosphorylated tau tangles are all associated with neuronal impairment in AD [3]. It has been noted that the structure and operation of oligomeric forms of Aβ and tau are among the most persuasive for comprehending pathogenesis, defining treatments, and providing a speculative synthesis for how their actions both start and fuel pathogenesis [4].

Numerous factors have been linked to the development of AD such as an abnormal blood–brain barrier (BBB), mitochondrial dysfunction, oxidative stress, metal ion imbalance, genetic risk factors, and more, as shown in Figure 1 [5,6,7,8,9]. The BBB serves a dual purpose, which involves the transportation of critical nutrients, hormones, and drugs to the brain while exporting metabolic end products from the CNS (as a carrier function). Additionally, it prevents circulating toxic substances and proteins with osmotic activity from entering the CNS (as a barrier function). As individuals age, the BBB stability decreases, causing its leakage. AD worsens this failure of BBB integrity, allowing peripheral immune cells to enter the brain, and potentially exacerbates the pathology by promoting harmful neuroinflammation. According to most evidence, Aβ buildup is what causes BBB damage to spread. Nevertheless, BBB damage is also seen in tauopathies without Aβ pathology, pointing to a potential function for tau in BBB damage [5,10,11]. Traditionally, the BBB was thought to block the entry of both cellular and humoral components of the immune system from the blood in the brain’s immunological privilege model. However, it is now increasingly clear that T cells frequently visit the brain. Although most T cells in the AD brain are not fully developed effectors, the cytokines they release may influence the pathogenic developments in AD [9,12]. Although T cell proliferation in the AD brain is low, once within the brain, T cells interact with microglia or astrocytes functioning as antigen-presenting cells (APC) to perform their effector tasks [13,14]. It is clear that microglia play a significant role in AD development. Early Aβ and tau deposition results in microglial activation, NLRP3 inflammasome assembly, and cytokine and protein release. However, because the original triggers, such as Aβ and tau, are not eliminated, continuous microglial activation exacerbates AD pathogenesis and results in greater protein buildup and neuroinflammation [14,15,16].

According to various studies, oxidative stress may increase the generation and aggregation of Aβ and facilitate the polymerization and phosphorylation of tau, thus creating a vicious cycle that favors the onset and progression of AD [17,18,19,20]. During the onset and development of AD, mitochondrial dysfunction is likely responsible for the initiation and/or amplification of oxidative stress. Oxidative stress can harm the integrity and function of the mitochondria. Defects in mitochondrial dynamics can result from metabolic or environmental changes, as well as responses to genetic deficiencies. This can limit the mitochondria’s ability to adapt to changing cellular needs, which may have notably detrimental effects on neurons [21].

It has recently been demonstrated that the strongest genetic risk factor for AD, the apolipoprotein E (*APOE*) *ε4* allele, is associated with a greater inflammatory response, although the exact mechanism is still unclear [22]. In the human population, there are three prevalent *APOE* alleles (*ε2*, *ε3*, and *ε4*), with *ε4* being a risk factor and *ε2* seen as protective compared to *ε3* [23]. For the brain to continue to function normally, metal ions including iron, copper, zinc, and calcium must remain in a state of homeostasis. The imbalance of these metal ions in the brain plays a key role in the development of AD. Incorrect deposition of iron, copper, zinc, or calcium in various brain locations can encourage tau hyperphosphorylation, excessive Aβ formation, and their accumulation [24].

This review covers the most prevalent changes related to AD. The purpose of this review is to provide an in-depth analysis of the mechanisms underpinning the onset and progression of AD. This paper presents the latest discoveries and updates on the various factors associated with AD, such as the role of T cells and microglia, genetic risk factors, and metal ion imbalance. Additionally, this review article highlights the role of heme in the development of AD and summarizes some mathematical models that demonstrate the effect of Aβ on AD. By consolidating these findings, this review seeks to expand our knowledge of AD and inform further research in this field, with the ultimate goal of identifying new therapeutic targets and developing effective treatments for AD. Section 2 discusses the neuropathological hallmarks of AD. One specific alteration that can play a pivotal role in the development of AD is the alteration of the BBB, which results in an increased occurrence of T cells in the brain. Section 3 reviews the role of the immune system in AD, while genetic risk factors associated with AD are presented in Section 4. The imbalances of metal ions such as heme, Fe, Cu, and Zn are discussed in Section 5. Section 6 focuses on mitochondrial dysfunction and oxidative stress, followed by altered signaling pathways in Section 7. Section 8 summarizes mathematical models of AD, including the stochastic and discrete deterministic models. The review is concluded in Section 9.

## 2. Neuropathological Hallmarks of AD

### 2.1. The Role of Aβ and Neurofibrillary tau Tangle in AD

The proteins Aβ and tau have been identified as key contributors to the pathophysiology of AD, largely due to their deposition in the histopathological brain lesions, the senile plaques for Aβ and the neurofibrillary tangles (NFTs) for tau. The soluble forms of Aβ and tau are also found to be increased in the brains of AD patients [25]. In healthy individuals, Aβ is naturally produced and eliminated from the brain at rates of 7.6% and 8.3% of total Aβ every hour, respectively. However, in late-onset AD (LOAD) this percentage is reduced by around 30% [26,27,28]. As a result of Aβ buildup, microglia and astrocytes are activated as part of the inflammatory response in an attempt to remove the plaque [3,29], but this also harms the nearby neurons and neurites. Additionally, NFTs, which are normally involved in intracellular activity, obstruct typical axonal transport, and eventually lead to neuronal death [3].

#### 2.1.1. The Effects of Aβ in AD

Mitochondria are the main energy source for brain cells to function properly. Aβ and amyloid precursor protein (APP) have been found in the membranes of mitochondria, where they interact with mitochondrial proteins, increase the formation of reactive oxygen species (ROS), and damage the structure and function of mitochondria. This can lead to disruption in normal neural function. Aβ oligomers can also harm mitochondria by causing an increase in intracellular Ca^2+^ levels and promoting the entry of Ca^2+^ into mitochondria, which can further damage their structure and function [30,31,32,33,34]. Cognitive impairments correlate with synaptic damage in AD [35]. It has been suggested that mitochondrial dysfunction and Aβ buildup at synapses can cause synaptic injury, impaired neurotransmission, and cognitive decline in aging and AD patients [36]. Malfunctioning mitochondria have been shown to increase Aβ generation from APP, and Aβ causes mitochondrial dysfunction [37]. P-glycoprotein (P-gp) is essential for the regular clearance of Aβ across the BBB and plays a crucial gatekeeping role. Endogenous Aβ peptide release from the brain is part of this process. Emerging evidence supports the hypothesis that defective P-gp activity promotes Aβ accumulation and contributes to the pathophysiology of AD [26,38]. Aβ oligomers are known to promote inflammation and oxidative stress. Moreover, there is evidence suggesting that inflammation and oxidative stress can also contribute to the formation of Aβ oligomers [20,26,39]. It has been proposed that the combination of anti-inflammatory and anti-oxidant drugs may be a useful strategy for treating AD [40]. Heme, a key functional form of iron in cells that is synthesized in mitochondria, binds to Aβ to form the Aβ–heme complex [41], which inhibits Aβ accumulation and results in heme deficiency. Heme shortage reduces mitochondrial complex IV’s activity and protein content, causing oxidative stress and disrupting Ca^2+^ homeostasis. Heme deficiency also affects zinc and iron homeostasis, APP, mitochondrial complex IV, and NO synthase (NOS) [42,43,44]. The aging brain exhibits many of the same phenotypic changes as heme-deficient cells, and these changes are more pronounced in neurodegenerative diseases such as AD. Heme-deficient brain cells cannot differentiate or conduct a complete cell cycle, suggesting that heme has a unique function beyond its traditional role in cell biology [44]. Table 1 summarizes the effects of Aβ.

#### 2.1.2. The Effects of tau in AD

The biological activities of the phosphoprotein tau are controlled by the amount of its phosphorylation. In the AD brain, tau is hyperphosphorylated [52]. NFTs, built up from hyperphosphorylated tau, are associated with tauopathies. Hyperphosphorylation causes tau to lose its common physiological role, become more toxic, and aggregate to form NFTs [48]. In previous studies, in situ hybridization was used to examine the distribution of α-tubulin mRNA in the human hippocampus of normal subjects and those with AD. The hybridization signal was significantly lower in NFT-rich regions, and NFT-containing neurons had a weaker hybridization signal than NFT-free neighboring neurons [53,54]. NFT-bearing neurons showed lower tubulin transcription, which might have a role in the reduced number of microtubules in these cells [54]. Microtubules are arranged in paraxial rows on the axons and dendrites. The structural backbone provided by the microtubule arrays allows axons and dendrites to develop and maintain their specific morphologies [55]. The development of NFTs is believed to be associated with neuronal dysfunction in AD. Microtubules are essential for maintaining the neuron’s shape. The abnormal phosphorylation of tau likely causes microtubule disruption by reducing the levels of functional tau. Acetylated α-tubulin immunoreactivity decreased in most NFT-bearing neurons, even in the neuronal population with a relatively lower tau immunoreactivity, indicating early microtubule instability [49,56,57]. It has been demonstrated that cells with tau overexpression are more susceptible to oxidative stress, and evidence suggests that oxidative stress may contribute to tau pathology [58]. Tau tangles, which are formed due to microtubule instability, consist of tau oligomers and monomers. The accumulation of tau tangles is linked to neuroinflammation, impaired synaptic function, dysfunctional autophagy, and dysfunctional mitochondria, all of which can cause neuronal injury. Additionally, tau oligomers can spread from one neuron to another [50,59,60,61]. Calafate et al. [62] used a mouse model to study the propagation of tau pathology, and found that the presence of synaptic contacts can facilitate tau pathology propagation between neurons and the amount of total accumulated tau significantly decreased when synaptic connectivity was reduced. Similarly, Wu et al. [63] used a mouse model to investigate how tau pathology spreads through the brain, finding that neuronal activity in one area of the brain can enhance the propagation of tau pathology to connected regions through a trans-synaptic mechanism. Overall, both studies provide evidence that tau oligomers can pass from neuron to neuron through a process of trans-synaptic spread. It has been demonstrated that the hyperphosphorylated tau that makes up the NFTs can hinder transport of mitochondria. This leads to an energy shortage and oxidative stress at the synapses, which can eventually lead to neurodegeneration [51]. A summary of the effects of tau can be seen in Table 1.

#### 2.1.3. Aβ and tau Interplay 

While Aβ and tau cause toxicity via different pathways, in vitro and in vivo research suggests that they can interact in three different ways [45,46,64,65]. The first mode of interaction is that Aβ drives tau pathology. In APP transgenic mice, Aβ deposition leads to hyperphosphorylated tau, whereas tau transgenic mice do not have observable Aβ plaque pathology [45,46]. It has been reported that mitochondrial malfunction and a lack of energy are early signs of AD [66,67]. In the second mode of interaction, both Aβ and tau contribute to interactive toxicity, damaging mitochondrial respiration in triple transgenic mice (triple AD mice), which show combined Aβ and tau pathologies. This results in an amplification of mitochondrial dysfunction when both pathologies are present together in mice compared to mice overexpressing tau or APP alone [46,64]. In previous studies [64,68], there was a considerable disturbance in the regulation of 24 proteins, out of which about a third were found to be mitochondrial proteins that are mainly associated with complexes I and IV of the oxidative phosphorylation system (OXPHOS). At both the protein and activity levels, the deregulation of complex IV was dependent on Aβ, whereas the deregulation of complex I was dependent on tau According to Vossel et al. [65], in the third mode of interaction, Aβ disrupts axonal transport, which is essential for neuronal activity. Aβ-induced axonal transport impairments are prevented by a decrease in tau. 

### 2.2. Neuroinflammation 

Another underlying mechanism of AD pathology is neuroinflammation [69,70,71,72]. Inflammation is essential for repair processes in the brain, but prolonged inflammation can impair brain function [73,74,75]. The molecular mechanisms underlying the progression from chronic, low-grade systemic inflammation to neurodegeneration are still not fully understood [76]. Neuroinflammation is most likely the result of AD pathologies and risk factors, and it exacerbates the disease severity [77]. AD brains show elevated levels of proinflammatory cytokines and inflammatory markers, possibly as a response to the Aβ plaques and NFT deposition, which induce neuronal damage or death [76,78]. Aβ deposition activates the complement system, microglia, and astrocytes, and induces the secretion of inflammatory mediators such as IL-1α, IL-1β, IL-6, and TNF-α, and reactive oxygen and nitrogen species, which lower phagocytosis and prolong neuroinflammation [77,78]. Proinflammatory mediators activate microglia during the AD pre-symptomatic stage, leading to synaptic dysfunction and neuronal death [77]. This implies that neuroinflammation is an early event in the development of AD pathology [73,79]. Moreover, in a previous study, activated microglia surrounding amyloid plaques and elevated proinflammatory cytokine levels were found in both the periphery and central nervous system (CNS), supporting the role of inflammation in AD [73,77]. Therefore, the inhibition of neuroinflammation could be a promising strategy to treat AD.

The role of cytokines in neuroinflammation in AD is diverse. TNF-α and IL-1 elevate the synthesis of Aβ from APP by β- and γ-secretase [80,81]. IL-1 also elevates tau phosphorylation by the p38-MAPK pathway [81,82]. IL-1β suppresses astrocytic sonic hedgehog production, downregulates tight junction proteins, and elevates astrocytic activation with subsequent pro-inflammatory cytokine production, BBB disruption, and neuroinflammation. IL-6 increases APP expression and tau phosphorylation by the cdk5/p35 pathway [78,83]. Furthermore, elevated levels of chemokines and cytokines in AD brains may attract circulating immune cells in the periphery to cross the BBB to the CNS, exacerbating the inflammation [84]. Chemokines can also attract microglia to the periphery of Aβ plaques. Elevated expression levels of chemokine receptors on activated microglia were found in the brains of AD patients. Additionally, increased levels of MCP-1/CCL2 and CCL11 chemokines may reflect pathological changes and memory function alterations found in patients with early AD [85]. On the other hand, SDF-1/CXCR4 chemokine can activate microglia and reduce Aβ deposition. SDF-1 levels are low in early AD patients and negatively correlate with tau protein levels in the cerebrospinal fluid (CSF), consistent with its neuroprotective function [79,86]. Some other chemokines found to be elevated in the blood of AD patients are IP-10, IL-13, IL-8, MIP-1α, and fractalkine; however, RANTES levels were found to be decreased in a previous study [79]. Nevertheless, controversy in the literature about these chemokines indicates that further studies are required to comprehend their role in AD progression.

A potential biomarker, the translocator protein (TSPO), an outer membrane mitochondrial protein, has been associated with neuroinflammation in AD [87,88,89]. TSPO expression is elevated in the brain periphery and is directly proportional to microglia, and potentially astrocyte, activation. TSPO radiotracers have been used for neuroinflammation imaging in vivo [73,90]. Other proteins that have been found to be dysregulated in AD are CSF1R, COX-1, COX-2, CB_2_R, P2X7, and P2Y12 receptors [91,92,93,94,95,96,97,98,99,100,101]. Radiotracers for these proteins have been developed; however, further studies are required to determine their utilization as potential imaging biomarkers to detect neuroinflammation [90]. Another inflammatory biomarker explored is the complement system. Lower levels of plasma C3, the central component of the complement system activation, are associated with a higher risk of AD. C3 and C4 levels were to be found increased in AD patients in previous studies [79,102,103]. Moreover, the acute-phase inflammatory protein C-reactive protein (CRP) may promote Aβ_42_ synthesis and the activation of the complement system in the AD brain. While some studies found elevated levels of serum CRP in AD brains, others found decreased levels [104,105,106]. This controversy is likely due to the changes in its levels as the disease progresses [79]. Targeting inflammation biomarkers can be exploited to image neuroinflammation and monitor AD progression in vivo, and may be a promising strategy for early diagnosis and therapeutic intervention.

There is evidence of an association between diet and AD development. The Western diet (WD), which is based on ultra-processed foods and is rich in carbohydrates, salt, fat, and cholesterol [107], can enhance brain amyloid accumulation, tau protein phosphorylation, systemic inflammation, and the impairment of memory, learning, and cognitive functions [76,108,109,110,111,112]. WD has also been linked to neuroinflammation. Chronic WD-fed AD mouse models showed elevated levels of activated macroglia and astroglial cells in the hippocampus and entorhinal cortex. Moreover, neuroinflammation was indicated by increased levels of proinflammatory genes (*Trem2, Treml2, Tyrobp, CX3CR1, Ccl3*) and markers of phagocytic microglial cells (Iba1, CD68, TREM2) around the Aβ plaques [113,114,115]. WD-induced neuroinflammation seems to occur earlier, before Aβ plaque formation and brain deposition. Neuroinflammation can occur as a result of impaired amyloid clearance due to a loss of microglial phagocytic function and astrocyte-dependent disruption of the glymphatic system. Additionally, a higher impact of WD on AD development was observed in APOE4 carriers in a previous study [76].

Several studies demonstrated a link between diet, obesity, and AD [76,116,117,118,119,120,121,122]. Obesity increases the risk of cognitive decline and AD development six-fold in adulthood and midlife [76,123]. Obese individuals have an increased brain atrophy rate, decreased cortical and hippocampal volume, low performance in memory tasks, and deficits in executive functioning [124,125,126,127,128]. Diet-induced obesity has been associated with increased APP, p-tau levels, and hippocampal Aβ, decreased hippocampal neurogenesis, and cognitive tasks deficit in AD animal models [129,130,131]. Diet-induced hypercholesterolemia experiments in mice showed the exacerbated neuroinflammatory response, increased p-tau levels, and the generation and deposition of toxic Aβ in neurons and astrocytes, leading to impaired cognition [132,133,134]. Other studies have demonstrated that diet-related dysbiosis and alteration of the gut microbiome composition in humans might disturb neurotransmitter production, disrupt synaptogenesis, contribute to systemic inflammation, impair the BBB, and contribute to cognitive impairment and AD development [76,135,136]. Furthermore, diverse studies have suggested that type 2 diabetes mellitus (T2DM) is a major vascular risk factor, and contributes to AD development [117,137,138,139,140]. Patients with T2DM have a two-fold higher risk of developing AD than healthy individuals [116,141,142,143]. Future studies should address the influence and interaction of environmental and genetic risk factors in AD development and progression.

### 2.3. BBB Alteration

The BBB is a complex structure composed of endothelial cells, pericytes, glial cells, and neurons, which form microvascular networks within the CNS [76]. In addition to protecting the brain, the BBB can regulate the blood-to-brain transport of nutrients, respond to soluble factors and plasma proteins, communicate with peripheral immune system cells, impede the entry of circulating cells and harmful molecules such as pro-inflammatory factors and toxins, and remove neurotoxic molecules and metabolic waste from the brain [76,79].

The loss of BBB integrity combined with the migration of immune cells into brain vessels may exacerbate inflammation and neurodegeneration in patients with AD [116]. Some pathological features of BBB breakdown in AD brains include the extravasation of blood-derived proteins (fibrinogen, thrombin, plasminogen, immunoglobin C, albumin) in the hippocampus and cortex, an increase in the CSF levels of a pericyte injury marker, an increase in the CSF-to-plasma-albumin ratio, iron accumulation, and brain microbleeds [76,117]. 

Diverse studies demonstrated that increased Aβ production and deposition induce BBB disruption [144,145,146,147]. Disfunction of the BBB impairs Aβ clearance, and can promote or escalate Aβ production [144]. Thus, a vicious cycle between BBB damage and Aβ accumulation can be established, resulting in neuronal damage and a loss of neuronal networks [144,148]. BBB dysfunction can also induce tau hyperphosphorylation, and tau pathology can trigger BBB damage [5,144,149,150]. Moreover, BBB permeability was found to be increased in APOE4 carriers [151,152,153,154], which show accelerated pericyte degeneration and BBB breakdown [155]. The loss of pericytes in AD is associated with fibrinogen leakage, reduced oxygenation, and fibrillar Aβ accumulation [156]. Additionally, astrocytic dysfunction and reactive astrocytes may induce increased generation of Aβ [144,157]. Another common event in AD is the loss of tight junctions, which is associated with insoluble Aβ_40_ and synaptic dysfunction. Cortical tight junction proteins such as claudin-5 and occludin have been found to be decreased in AD brains [158]. Targeting the BBB and its components can result in a promising strategy to treat AD. 

### 2.4. Cholinergic Neurons in AD

Cholinergic neurons are nerve cells that secrete the major brain neurotransmitter acetylcholine [159]. Early studies in rodent models and humans delineated the importance of acetylcholine and cholinergic neurons in memory function [160,161]. Although these neurons are localized in specific regions of the brain, they project into almost all brain regions and have been implicated in diverse behavioral, cognitive, and systemic functions [162,163,164,165]. Acetylcholine released by the cholinergic system modulates behavioral flexibility [166,167,168], attention [169,170], and arousal [171,172,173]. Of particular interest are the basal forebrain cholinergic neurons, which are essential for cognitive and memory function [174]. These basal forebrain cholinergic neurons form the major projection into the cerebral cortex and hippocampus [175,176,177,178]. These central cholinergic neurons display age-related dysfunction, leading to mild cognitive impairment [179,180,181].

The selective reduction in pre-synaptic cholinergic enzyme activity in specific brain regions of AD patients was first reported in the 1970s [182]. This finding pointed to a possible role for the loss of cholinergic neurons in the onset of AD. It was then shown that this cholinergic system failure is due to the degeneration of cholinergic neurons in the basal forebrain region responsible for the innervation of the cerebral cortex [183,184,185]. This gave rise to the concept of the “cholinergic hypothesis’’, which posited the loss of cholinergic neuronal function and acetylcholine activity as the primary event in the onset of AD [186,187]. This hypothesis was further strengthened after it was shown that treatment with an acetylcholinesterase inhibitor, tetrahydroaminoacridine (THA, Tacrine), produced clinical improvements in AD patients [188]. Although tacrine was the first acetylcholinesterase inhibitor approved by the Food and Drug Administration (FDA) for the treatment of AD and provided modest improvement in cognitive and memory symptoms in mild AD patients, it was withdrawn due to its severe hepatotoxicity [189,190].

Acetylcholine impairment is the most widespread and important neurotransmitter alteration in AD [191]. Acetylcholinesterase hydrolyses acetylcholine to choline and acetate, and is an essential enzyme of the neuronal cholinergic system [192,193]. Decreased acetylcholine availability resulting from cholinergic deficiency can be corrected by targeting acetylcholinesterase via inhibition. The goal of this approach is to increase the neuronal availability of acetylcholine and cholinergic neurotransmission, and make up for the shortfall in secretion and synaptic availability of acetylcholine resulting from central cholinergic neurons’ dysfunction and degeneration in AD. Thus, following in the footsteps of Tacrine, several reversible inhibitors have been developed for the treatment of AD. Donepezil, galantamine, and rivastigmine are acetylcholinesterase inhibitors that have been approved by both the Food and Drug Administration and the European Medicines Agency for the treatment of AD [194,195]. In addition to these three, only two other treatments (Memantine and Aducanumab) have been approved for AD [196]. These drugs elicit improvement in certain cognitive and behavioral symptoms in mild to moderate AD, but do not treat the underlying problem of cholinergic neuronal degeneration or prevent disease progression [197,198]. In addition, they have several drawbacks such as (1) low efficacy, which wanes further as the disease progresses, and (2) severe adverse side effects such as cardiac arrhythmia, nausea, diarrhea, vomiting, and muscle cramps [199,200,201].

There is ongoing research to develop more effective and less toxic cholinergic inhibitors, as well as to incorporate molecular features of acetylcholinesterase inhibitors in multitarget drugs [202,203]. Drugs that improve cholinergic function are among the few FDA-approved therapeutics to treat AD, and will remain indispensable tools in the multifaceted approach to the treatment and management of AD patients. Therapeutics aimed at improving the cholinergic system have not been widely researched in recent years as attention has shifted to other promising mechanisms underpinning AD. Developing new therapeutics that can improve cholinergic function can provide further ammunition in the continued search for effective AD treatments.

Basal forebrain cholinergic neurons (BFCN) depend on nerve growth factor (NGF) for the maintenance of their biochemical phenotype. NGF released by cortical and hippocampal neurons also regulates the BFCN synaptic integrity and number [204]. The dependence of BFCN on NGF informed the hypothesis that BFCN atrophy is caused by NGF deficiency. Interestingly, the biosynthesis of NGF in the cerebral cortex is not altered in AD [205], while there is an increase in the levels of its precursor, proNGF [206,207]. However, evidence has implicated an altered NGF signaling system and interaction with BFCN in AD [208,209]. Dysfunction in the NGF pathway seemed to be due to impaired retrograde transport and maturation of NGF [210]. Genetic delivery of NGF to the brain of AD patients has been used to activate the neuronal trophic response [211]. NGF-based therapy seeking to boost NGF trophic activity should be explored to halt BFCN atrophy, especially in early stages of AD.

### 2.5. Glial Cells 

The nervous system is composed of two major cell types: neurons and glial cells, which include astrocytes, microglia, oligodendrocytes, polydendrocytes (NG2 glia), and Schwann cells. Astrocytes play a vital role in regulating neurogenesis and synaptogenesis, shaping the micro-architecture, providing structural support for neurons, and protecting the brain from injuries [212,213,214,215,216]. Microglia cells are the major immune cells of the central nervous system and serve as resident macrophages of brain parenchyma, providing the first line of defense against external insults [217,218,219,220]. Microglia are also involved in synaptic formation and remodeling [221,222,223]. As part of the complement cascade of the CNS innate immune system, macroglia engulf and prune synapses [224,225,226]. Oligodendrocytes are the CNS cells responsible for myelin formation. They wrap myelin sheaths around the neuronal axon and ensure the proper propagation of the electrical action potential through the axon [227,228]. NG2 glia (also called oligodendrocyte precursor cells) are used to produce oligodendrocytes during development and adulthood, and they are also involved in the maintenance of microglia homeostasis and the regulation of brain innate immunity [229,230,231,232,233]. The essential functions of glial cells in normal brain development, function, immune response, and homeostasis mean that disruption of normal glial function can cause neuro-glial metabolic dysfunction, which can lead to and accelerate neurodegeneration. Evidence has linked impaired glial cell function and homeostasis to the pathogenesis of AD, as discussed below.

#### 2.5.1. Astrocytes

Astrocytes are the most prevalent glial cells, and constitute about a third of cells in the human central nervous system [214,215]. Many mechanisms link astrocytes to the development and progression of AD. *APOE*, the major genetic risk factor for LOAD, is expressed in the astrocytes of the normal brain. Astrocytes exhibit reactive astrogliosis in response to a CNS infection or injury. Reactive astrogliosis is the functional, molecular, cellular, morphological, and population alterations in astrocytes that serve as a protective mechanism against insults [234,235,236,237]. Reactive astrogliosis has been reported in the brains of mouse models of AD and AD patients. Astrocytes become more reactive and associate with Aβ plaques in the AD brain [236,238,239]. The reactive astrocytes display morphological hypertrophy and increase the expression of glial fibrillary acidic protein (GFAP). GFAP is being explored as an early marker of AD with promising results [240,241,242]. Aβ can activate astrocyte reactivity, upregulate the release of pro-inflammatory cytokines [243,244,245,246], and alter astrocyte-Ca^2+^ homeostasis [247,248,249]. Impaired calcium signaling mediated by astrocytes, especially increased Ca^2+^ activity leading to the hyperactivation of Ca^2+^-dependent proteins, can promote the release of transmitters such as GABA, which alters neuron-glial communication, disrupt synaptic plasticity and transmission, and accelerate cognitive deficits [247,250,251].

Atrophic astrocytes have been documented in both mouse models of AD and the post-mortem brains of AD patients [252,253,254]. It has been suggested that atrophy of astrocytes might be an early event in AD, leading to the loss of astrocyte homeostasis and function. Recent evidence indicates that astrocyte dysfunction and disease-associated astrocytes are present in the early stages of AD, providing support for this claim [253,255]. Astrocyte dysfunction can lead to synaptic impairment, which produces cognitive deficits that are hallmarks of the early stages of AD. Astrocytes are involved in the phagocytosis and clearance of Aβ [256,257]. A defective astrocyte might not be able to effectively clear Aβ. In the later stages of neurodegeneration, as Aβ deposition increases, senile plaques promote the formation of reactive astrocytes, which, in turn, activate microglia. Activated microglia release neurotoxic and neuroinflammatory factors, which induce synaptic impairment, neuronal loss, and severe neurodegeneration in a feedforward vicious cycle.

#### 2.5.2. Microglia

Microglia are innate immune cells of the CNS, and are vital for normal brain development and function both in the healthy and pathological state. They control the immune response in the brain by interacting with other immune cells [258,259,260]. Microglia are transformed into reactive states in response to infections and injuries in a process known as microglia activation. Depending on the type of stress or pathological change, reactive microglia undergo specific changes in phenotype, morphology, proliferation patterns, and activity. They release various neurochemicals including growth factors, cytokines, chemokines, and other inflammatory mediators [258,261,262]. Reactive microglia are a neuropathological hallmark of AD brain response to amyloid deposition, neurofibrillary tau tangles, and neuronal death [263,264,265]. Reactive microglia are widely considered to be a manifestation of AD pathology, but recent evidence points to active roles for microglia in the pathogenesis of AD and their potential as a therapeutic target.

#### 2.5.3. AD Risk Loci Are Expressed by Microglia

Several genetic loci, including *APOE, TREM2, SPI1,* and *CD33*, which are highly associated with the risk of AD, are predominantly or exclusively expressed by microglia [266]. The *APOE4* allele is a major genetic risk factor for LOAD [267]. *APOE* is involved in amyloid clearance and plaque formation. The *APOE4* allele seems to reduce amyloid removal and elevate amyloid deposition in plaques [268,269,270]. *APOE4* has been reported to alter lipid homeostasis in glia cells [271] and promote dysfunctional microglia phenotypes in neurodegeneration [272]. TREM2 is a surface receptor expressed on myeloid cells including microglia. TREM2 is involved in the clearance of Aβ, the removal of cell debris, and microglial survival [273,274,275,276]. Dysfunctional TREM due to mutations in the *TREM* gene is likely to contribute to an increased risk of AD, although the exact mechanisms are not fully understood.

#### 2.5.4. Microglia in the Complement System and Inflammatory Signaling

The microglia-mediated innate immune complement system is essential for engulfing and shaping synapses during development [277]. Some genes, such as *CR1*, involved in the complement system, are genetic risk factors for AD development [278,279]. Hyperactivation of the complement system is possibly involved in synaptic loss and neuronal degeneration in AD. This hypothesis is supported by experiments that show that inhibition of the complement system protects against synaptic loss and neurodegeneration [224,278].

### 2.6. Synaptic and Neuronal Loss in AD 

A progressive decline in cognitive function is a major symptom and hallmark of AD [280,281]. Cognitive deficit has been reported to be present at early stages of the disease and decades before amyloid deposition [282,283]. Multiple lines of evidence point to synapse loss as the major cause of cognitive impairment in AD. Examination of the postmortem brains of AD patients shows that synapse loss is the best correlate to cognitive deficits in AD [284,285,286,287,288,289]. Synaptic dysfunction and loss have been extensively reported in animal models of AD [290,291,292,293,294]. The correlation between synapse loss and cognitive defects has been demonstrated using various methods and models [295,296,297,298].

Neuronal loss is a canonical neuropathological feature of AD [299,300,301]. The loss of neurons starts at early stages and accelerates as the disease progresses [302,303,304]. There is a pronounced loss of neurons in the entorhinal cortex and hippocampus compared to other brain regions [303,304,305]. Neuronal loss in these regions closely correlates with cognitive impairments [306,307]. Various AD transgenic mouse models replicate the neuronal loss seen in AD [308,309,310,311,312].

#### Mechanisms of Synaptic Loss and Neuronal Loss

Aβ is connected to synaptic dysfunction in AD. However, it is the soluble oligomeric Aβ species, and not the total amyloid plaque deposition, that constitutes the most toxic forms of amyloid, contributing to synaptic damage and neuronal loss in the disease. [313,314,315,316,317]. The mechanism of Aβ-induced damage possibly involves the binding of Aβ to synaptic-associated proteins [318,319]. Advanced imaging techniques have been used to show the presence of soluble amyloid species in synapses [318,320]. The AD generic risk gene *APOE4* promotes the transport of these amyloid species to the synapse, where they exert toxic effects and mediate neuronal loss [321].

The accumulation of tau proteins correlates with cognitive decline and clinical symptoms in AD [322]. Although tau is found in healthy and AD synapses, there are higher levels of hyperphosphorylated tau in the synapses of the AD brain [323,324,325,326,327]. Tau toxicity to synapses and neurons seems to be mediated by soluble, hyperphosphorylated, oligomer forms [328,329,330,331]. Normal tau protein is involved in the axonal transport of mitochondria and other important molecules to synapses [332]. Impaired axonal transport resulting from tau pathology can contribute to synaptic dysfunction and neuronal atrophy. Indeed, reduced mitochondrial numbers have been observed in AD brains [65,326,333,334].

Inflammation, both local and systemic, possibly contributes to synaptic loss and neuronal death. Glia cells are highly involved in the innate immune system-mediated inflammatory response through the release of various factors and the expression of proteins that can affect synaptic function [29,335]. Microglia functions such as the pruning of synapses via the complement system, release of pro-inflammatory cytokines, and phagocytic activities have direct effects on synaptic integrity [336,337].

Neuronal cell death in AD has been reported to be mediated by cellular processes involving apoptosis, necrosis, autophagy, parthanatosis, ferroptosis, pyroptosis, and the mitochondrial permeability transition pore [338].

## 3. The Role of Immune System in AD

The BBB plays a critical role in maintaining a stable and favorable environment for neurons to function properly. It does this not only by physically separating the blood from the brain, but also by expressing particular transporters and ion channels that maintain the proper ion balance required for appropriate synaptic function [3,339]. The BBB is essential for maintaining the homeostasis of the CNS. BBB dysfunction is associated with a number of CNS disorders, including AD. The BBB impairment in AD can be both a source and a result of Aβ buildup in the CNS [340,341,342]. The BBB is highly durable, preserving its integrity in the face of significant brain atrophy, neuroinflammation, and toxic tau buildup. However, it has been shown that tau itself can trigger disruption of the BBB. Additionally, when tau levels drop, the BBB might regain its integrity. Moreover, Aβ buildup, especially along the vasculature, is responsible for this rise in BBB impairment [5,343].

### 3.1. The Role of the Innate Immune System in AD

In AD, activated microglia are found in the direct vicinity of Aβ plaques. The innate immune system in the brain is made from microglia. The microglial innate immune system is activated by Aβ and tau proteins contained in the AD brain, but because they accumulate over time rather than being eliminated from the microglia environment, this leads to chronic microglial activation [14,344]. Triggering receptor expressed on myeloid cells 2 (TREM2), as a microglial Aβ receptor, binds to Aβ oligomers and mediates Aβ degradation and activation in microglia [345]. Regulating the brain’s levels of Aβ is one of the most significant roles played by glial cells in AD. Microglia and astrocytes have been associated with this process as key moderators of Aβ elimination and degradation. In numerous circumstances, astrocytes and microglia have a crucial function in the phagocytosis of Aβ through a variety of receptors. Toll-like receptors (TLR) are engaged in monomeric, oligomeric, and fibrillar Aβ elimination in the microglia [346,347]. It has been revealed that pCD14, TLR4, and TLR2 are necessary at the cell surface for the recognition of fibrillar-amyloid β (fA) and the initiation of a phagocytic response [348]. 

### 3.2. Role of B Cells in AD

T and B cells constitute a substantial portion of the adaptive immune system [14]. Although B lymphocytes are not thought to penetrate the AD brain, evidence connects their existence in the brain parenchyma to the development of the illness. Immunoglobulin deposits around Aβ plaques are the consequence of the entry of B lymphocytes into the brain parenchyma. While it is believed that immunoglobulin promotes and facilitates the uptake of Aβ plaques by microglia, the Aβ-IgG complex in CSF is believed to adversely influence the cognitive condition of AD sufferers. Additionally, therapeutic B cell reduction, and thus IgG, slows the development of AD in mice, indicating that AD patients may also benefit from B cell targeting to control the disease’s progression [349,350,351]. Aβ or phosphorylated tau are the targets of immunoglobulins (Igs) generated by B cells, according to Dujardin et al. [22]. One technique to eliminate self-antigens such as Aβ is to use immunoglobulin M (IgM) antibodies. B1 cells, a subgroup of B cells, generate IgM antibodies. B1 cells are essential for the elimination of self-antigens without overt inflammation, and they spontaneously produce IgM during homeostasis [1,352]. While the role of B cells is typically thought to be advantageous, recent research on AD animal models has shown that a genetic loss of B cells alone or their depletion at the onset of AD may be beneficial. This highlights that B cells may have a notable detrimental impact on AD progression [353].

### 3.3. Role of T Cells in AD

T cells, one component of the adaptive immune system, develop in the thymus and become activated in the presence of pathogens [354]. When faced with an infectious exposure, T cells that are specific for the antigen proliferate and supply essential aid in the generation of cytokines or even in the direct destruction of infected host cells to clear the body of the bacteria or virus [14]. T cells have been demonstrated to cross the BBB and penetrate the CNS, and both the elderly human hippocampus and the AD brain contain them [355]. A study comparing the majority of AD cases to the control group showed that the number of T cells is higher in AD. T cells have been activated and are in the process of becoming effector T cells. Clonal expansion that is initiated by an antigen does not appear to occur. T cell emigration or survival, which leads to an aggregation of T cells in the brain parenchyma, may be influenced by the local inflammatory environment, even if the mechanisms that attract T cells to the AD brain are still not understood [9,356]. T cell extravasation is triggered by tau-related neurodegenerative alterations in the late stages of AD, according to Merlini et al. [357]. In a mouse model of tauopathy, hippocampal T cell infiltration induces neuroinflammation and cognitive loss, while T cell removal prevents cognitive deficits [358]. Elderly people and those with AD have higher levels of T cell reactivity to the Aβ. However, T cells are more common in the brains of AD patients than in those of people with other degenerative dementias or age-matched controls [359].

In AD, the number of regulatory T cells (Tregs), which limit immunological activity, decreases. In a previous study, Tregs from AD patients lacked their inhibitory ability and were not able to restrict the growth of CD4^+^ T cells. Treg cell depletion worsened cognitive impairment in APP/PS1 mice, while raising the Treg population restored cognitive function [14,360,361]. Tregs either secrete immunosuppressive anti-inflammatory cytokines, including TGF-β, IL-10, and IL-35, or they directly cause cytotoxicity and apoptosis by expressing granzyme B and perforin 1 to inhibit T helper cells’ activities [362].

Th1, Th2, and Th17 cells are produced as a result of the differentiation and proliferation of CD4^+^ helper T cells [363,364]. T cells have the ability to regulate the activity of their APC and surrounding cells through interaction and cytokine release. The production of pro-inflammatory cytokines by Th1 and Th17 cell types may have a negative bystander effect on nearby neurons or glial cells, exacerbating the inflammatory environment in AD. However, in a mouse model, the induction of Aβ-specific Th2 cells reduces cognitive issues and amyloid accumulation in blood vessels [354,365]. In the mouse brain parenchyma, CD8^+^ T cells have been found to be closely linked to both microglial and neuronal structures, and these cells may directly influence neuronal dysfunction in regulating synaptic plasticity. The phenotype of microglia may be influenced by CNS-specific CD8^+^ T cells, which may then cause neuroinflammation and, consequently, synaptic plasticity [366].

### 3.4. Role of Cytokines in AD

Inflammatory cytokines such as IL-1 and IL-6 play a role in the buildup of Aβ. It has been demonstrated that in response to the buildup of Aβ peptide, both microglia and astrocytes secrete IL-1, which, in turn, increases the production of Aβ, speeding up Aβ deposition. Additionally, it has been demonstrated that Aβ peptide raises IL-6 generation by astrocytes either directly or indirectly through an IL-1β-dependent pathway. Therefore, it appears that IL-6 acts as a secondary mechanism, enhancing the inflammatory reaction triggered by IL-1 [367,368,369,370]. Proinflammatory cytokines such as IL-1 and IL-6 are prevented from being produced in vitro by IL-10 cytokines [371]. Increased IL-10 levels also enhanced spatial learning in a previous study [372]. According to some research, IL-10 may be used therapeutically to treat neuroinflammation, cognitive decline, and neurodegeneration. In addition, limiting transforming growth factor-β signaling or administering Protollin as a nasal vaccine caused elevated IL-10 transcript levels together with decreased β-amyloidosis [372,373]. In a mouse model, IL-4 can lower levels of monomeric Aβ and several Aβ oligomeric forms. Decreased Aβ levels significantly support an advantageous impact of IL-4 on cognitive performance. It has been proven that IL-4 prevents mice from having defective spatial learning [374]. When compared with controls, AD patients were shown to have significantly higher IL-21 levels in a previous study [375]. The inflammatory cytokine IL-21 significantly correlated with IgG levels in AD, and it is possible that IL-21 contributed to the elevated IgG response [1]. The change in IgG in the mouse brain may be a crucial factor in promoting the buildup of Aβ and the progression of AD [376].

It has been claimed that Th17 cells, which have been found to enter the brain parenchyma of AD patients, contribute to the neuroinflammation and neurodegeneration of AD by secreting proinflammatory cytokines and by directly acting on neurons via the Fas/FasL apoptotic pathway. Th17 cells are very proinflammatory and create proinflammatory cytokines such as interleukin (IL)-17, IL-21, and IL-22. In individuals with AD, levels of the transcriptional factor (ROR γ) involved in the development of Th17 cells as well as the cytokines produced by Th17 cells are much higher. More information about the function and procedure of Th17 cells in the creation and progression of AD is still needed [375,377,378,379]. It has been established that interferon-*γ* (IFN-γ) and interleukin-4 (IL-4) are the two main cytokines secreted by Th1 and Th2 cells, respectively [380,381]. It has been proposed that an unmanaged increase in IL-4 or IFN-γ, as well as other pro- and anti-inflammatory cytokines, may enhance the negative effects of CNS damage or long-term neurodegeneration. It has been suggested that the duration of the reaction is most likely significant and that replacing an initial proinflammatory response, such as IFN- γ, with an anti-inflammatory response, such as IL-4, may remove a crucial beneficial impact of immune cells on astrocytes, namely keeping their glutamate clearance function [381]. In reaction to Aβ_1-42_(use the same designation as before?), IFN-γ causes an increase in neuronal death [382]. IL-4 and IL-10 are examples of anti-inflammatory cytokines that Th2 cells produce in contrast [383]. A summary of the effects of T cells and cytokines is shown in Figure 2.

## 4. Genetic Risk Factors Associated with AD

Early-onset or familial AD is linked to mutations in the amyloid precursor protein (*APP*), presenilin 1 (*PSEN1*), or presenilin 2 (*PSEN2*) genes involved in the amyloidogenic pathway [76,117]. APP is cleaved by β- and γ-secretase to produce Aβ [384,385]. Initially, the β-APP-site cleaving enzyme (BACE), a membrane-bound aspartyl protease, cleaves APP, generating a secreted APP derivative, sAPP, and the membrane-bound protein β-secretase-derived C-terminal fragment (CTFβ). Later, CTFβ is cleaved by γ-secretase, producing different lengths of Aβ, which are released into the extracellular space, and an APP intracellular domain (AICD) released to the cytoplasm [386,387]. Aβ_40_ is the most abundant form of Aβ produced into the cells. However, Aβ_42_ is the one that mostly mediates neurotoxicity. Aβ_42_ is more insoluble than Aβ_40_ and more likely to form aggregates [388]. Moreover, an elevated Aβ_42_/Aβ_40_ ratio is considered a pathogenic hallmark of AD [386]. Mutations in the *APP* gene usually affect the BACE and γ-secretase cleavage sites and the mid-domain Aβ region [386,389]. In total, 24 mutations were identified in *APP* and have been associated with increased Aβ_42_:Aβ_40_ in AD brains [117].

*PSEN1* and *PSEN2* are involved in the γ-secretase cleavage of APP to produce Aβ. Several *PSEN* mutations (185 mutations in *PSEN1* and 13 mutations in *PSEN2*) have been discovered in AD and have been implicated in the elevated levels of Aβ_42_:Aβ_40_ in the brain. *PSEN1* mutations have been also linked to disruption of cerebral blood vessels, degeneration of pericytes and mural cells, breakdown of the BBB, and Aβ deposits in small cerebral blood vessels [117]. A recent study re-evaluated genetic variations of the genes involved in FAD. A total of 288 *PSEN1* variants were classified as pathogenic/likely pathogenic variants, followed by 31 *APP* variants classified as pathogenic/likely pathogenic. The least pathogenic/likely pathogenic variants were from the *PSEN2* gene, with only 13 variants [390]. APOE is a protein that transports cholesterol and other lipids (apolipoprotein lipid) between cells in the CNS [389,391]. The main genetic risk factor for late-onset or sporadic AD is the *ε4* allele of *APOE*, one of the three possible alleles (*ε2, ε3,* or *ε4*) [76,117]. The APOE isoforms differ in two amino acid residues at positions 112 and 158, which can be either Arg or Cys. These polymorphisms cause differences in the lipid binding properties of the APOE isoforms and receptor affinities; APOE4 is hypolipidated compared to APOE3 and APOE2 [269,389,392] (Figure 3). Carrying the *ε2* allele reduces the risk for AD about 0.6-fold, while having the *ε3* form is believed to have a neutral effect on AD risk. Individuals who carry a copy of *ε4* allele have a risk of developing AD about three times higher than those with two copies of the *ε3* allele, while those with two copies of the *ε4* allele have an 8- to 12-fold higher risk. Additionally, those individuals who carry the ε4 allele are more likely to develop Aβ accumulation and AD dementia at a younger age than those carrying *ε2* or *ε3* alleles [76,389,391,393]. APOE4 increases pericyte degeneration, BBB breakdown, cerebral amyloid angiopathy, and fibrinogen deposits in the brain [117]. APOE binding and Aβ clearance capacity differ among isoforms. APOE2–Aβ, followed by the APOE3–Aβ complex, has a higher binding capacity to a major Aβ clearance receptor LRP1 at the abluminal side of the BBB, and is rapidly cleared across the BBB into circulation, whereas APOE4–Aβ complexes have a lower interaction with LRP1 and are cleared from the brain by a slower and less efficient mechanism that involves VLDLR. Additionally, APOE4 is associated with elevated vascular stiffness and increased vascular Aβ deposition; combined with a low CSF flow and low Aβ clearance, Aβ accumulation may escalate in the brain [76,117].

Another protein downregulated in the AD brain endothelium is PICALM [117]. Genome-wide association studies (GWAS) found an association between AD and SNPs present in the non-coding region of *PICALM.* PICALM is a protein involved in endocytosis, cell receptor internalization, and endocytic protein intracellular trafficking [394,395,396,397,398]. PICALM regulates cleaved APP C-terminal fragment degradation via autophagosomes and clathrin-mediated endocytosis of gamma-secretase [399]. PICALM is recruited as a result of the Aβ binding to LRP1 at the abluminal side of the BBB, and starts an immediate PICALM/clathrin-dependent endocytosis of Aβ-LRP1 complexes. Subsequently, PICALM continues to guide the trafficking of the endocytic vesicles across the brain endothelium. Later, it fuses with Rab5 and leads to Aβ intracellular transport. Finally, the binding of PICALM with Rab11 induces Aβ exocytosis at the luminal side of the BBB [117,400]. Low or deficient expression of PICALM is correlated with Aβ accumulation and low clearance across the BBB [400]. Another protein involved in Aβ clearance is clusterin (CLU). CLU binds to Aβ, prevents its aggregation, and induces its clearance across the BBB via the LRP2 pathway. GWAS found an association between *CLU* and sporadic AD, where the *C* allele is considered an AD risk factor. Rare coding variations in CLU in the β-chain were found to be enriched in AD patients [117,399,401,402].

A GWAS study also found an association between AD and the mesenchyme homeobox gene 2 (*MEOX2*), a transcription factor expressed in the brain vasculature and regulator of cell differentiation and remodeling [403]. AD brain endothelial cells express low levels of MEOX2, leading to vessel regression, LRP1 proteosomal degradation, and low Aβ clearance across the BBB [117,404]. Other mutations that influence AD risk are those on *ABCA7, BIN, CD2AP, PLD3, UNC5C*, and *AKAP9*. ABCA7 deficiency accelerates Aβ deposition in AD mouse models. In a previous study, *SORL1* was also found to be associated with AD. Mutations in the *SORL1* gene induced decreased binding and Aβ turnover. Furthermore, *MEF2C* and *PTK2B* have been associated with AD pathology. Additionally, mutations in genes involved in the immune system pathways have been associated with AD [399]. A GWAS revealed a connection between immune related mutations in the genes *TREM2, CD33, CR1, IL-8, ABI3*, and *PLCG2* with AD [73,79,399]. TREM2 is a receptor that binds phospholipids and APOE- and clusterin-containing lipoprotein particles, and promotes phagocytosis of lipoprotein particle–Aβ complexes [275,405,406]. Its expression is elevated in response to brain injury in resident microglia and infiltrating monocytes and macrophages in AD. The TREM2 missense variant (R47H) can increase AD risk by about two-fold [407,408,409]. CD33 is a receptor found on myeloid and microglia cells, and is involved in the anti-inflammatory immune response. Changes in CD33 expression affect Aβ levels. Monocytes expressing mutant CD33 showed elevated CD33 expression and reduced Aβ phagocytic capacity, which correlated with elevated levels of Aβ in the brain in a previous study [399,410]. The complement receptor 1 (*CR1*) is another high AD risk factor. CR1 inhibits complement activation through C3b and C4b in blood cells and microglia. The complement system is activated in AD, and research on mice treated with Aβ has reported elevated synapse elimination by phagocytic microglia. The association of AD risk with mutations in *CR1* can be explained by the overproduction of a longer CR1 isoform, which increases the number of C3b/C4b sites [278,398,399,411].

## 5. Metal Ion Imbalance in AD (heme, Fe, Cu, Zn)

Biometals play essential roles in normal physiology, and the association between their dysregulation and AD pathology has attracted growing research interest. An imbalance of metal ions may cause neurotoxicity and neurological damage. Elements including iron, calcium, copper, zinc, and selenium have been found to be involved in AD pathology [6].

### 5.1. Iron Imbalance

Iron is the most abundant metal in the brain, and participates in neurotransmitter synthesis, myelination, and mitochondrial function [6]. MRI and mass spectrometry studies have shown iron accumulation in the bilateral hippocampus, frontal and parietal cortex, frontal white matter, putamen, caudate nucleus, and dentate nucleus in AD patients [6,117]. *APOE4*, the main genetic risk factor for AD, has been reported to elevate iron levels [412,413,414]. Brain iron accumulation has been implicated in Aβ and hyperphosphorylated tau deposition, cognitive impairment, and brain atrophy, accelerating AD progression. Thus, iron may be considered a potential biomarker for AD progression. Increasing evidence shows that iron binds with Aβ and tau and accumulates with senile plaques and NFTs in neurons in AD brains [6,412,415,416]. 

Iron levels increase with aging and accumulate considerably in the brains of AD patients. Abnormal levels of iron transport (transferrin), iron storage (ferritin), and iron export (ferroportin) regulator molecules are associated with ferroptosis and AD [412,417,418,419,420,421,422]. Excess free iron escalates Aβ plaque formation, increases ferritin expression, promotes neuronal toxicity, and impairs spatial memory in AD mouse models [6,412]. An in vitro and in vivo study involving IFN-γ-induced microglia and AD mouse models revealed elevated iron accumulation, ferritin expression, and glycolysis, and decreased Aβ phagocytosis [423]. Transferrin variant C2 may be associated with an elevated risk of AD [412,424,425].

Another study reported a disturbance of the peripheral iron metabolism, where low levels of plasma iron, possibly due to deficient transferrin processing, may compromise hemoglobin production and lead to a reduction in hemoglobin levels in AD [6,426]. Other research found ferroportin and ceruloplasmin (a cellular iron transport mediator) proteins to be downregulated in AD [427,428,429]. Additionally, iron dyshomeostasis is associated with the main proteins involved in AD pathology. Iron modulates APOE regulation in neurons and astrocytes and increases its secretion [430]. Iron has also been involved in APP processing [431,432]. Moreover, iron binds to tau and facilitates its aggregation and phosphorylation. Tau protein mediates APP trafficking to the cell surface, where APP stabilizes ferroportin and promotes iron export [6]. It has been revealed that CSF ferritin and serum non-heme iron levels may be utilized as biomarkers to predict the progression of mild cognitive impairment (MCI) to AD [412,433,434,435]. Iron-related protein levels, such as ferritin, an iron-sequestering protein, and ferroportin1, an iron exporter, are reduced in AD pathology. Free Fe^3+^ induces aggregation of hyperphosphorylated tau. On the other hand, Fe^2+^ can reverse the aggregation [436]. Discrepancies in expression levels of iron transport-related proteins such as ferritin [412,436] may be due to the differences in samples utilized in the studies. Therefore, further studies involving populations from different races, sexes, genders, ages, and ethnic groups are required to elucidate iron’s role in AD pathology.

#### Heme Dysregulation

Heme (iron protoporphyrin IX) is a crucial co-factor for hemoglobin, cytochrome c, and cytochromes in complexes II, III, and IV of the mitochondrial electron transport chain (ETC). It is involved in diverse biological mechanisms such as oxygen and electron storage and transfer, including neuronal survival and differentiation. Heme excess promotes ROS production, leading to oxidative stress, lipid peroxidation, and cell death [437,438]. Studies in the brains of patients with AD showed that iron overload in the inferior temporal cortex may be involved in accelerated cognitive decline [436]. Dysregulation of heme metabolism and elevated heme/iron levels are associated with a poor prognosis in AD. Moreover, a heme molecule can bind an Aβ peptide at the N-terminus (H13) to generate Aβ–heme complexes. A second Aβ peptide may provide R5 and Y10 for the peroxidase activity of the complex, which is observed in presence of H_2_O_2_. However, the C-terminus of Aβ may also considerably contribute to the peroxidase activity of the complex [438,439]. Despite the growing evidence that highlights the importance of heme–Aβ interaction in AD pathology, the exact stoichiometry, structure, and biochemical mechanisms responsible for the peroxidase activity of these complexes are unknown and require further studies. Furthermore, disturbed heme metabolism and mitochondrial dysfunction of neuronal cells may contribute to Aβ–heme complex formation [41,439]. Aβ–heme complex-dependent peroxidase activity contributes to AD pathology, exacerbating oxidative stress and neurotoxicity, leading to neurotransmitter oxidation and Aβ aggregation. The peroxidase activity of the complex may result from the activation of cerebral immune cells that induce an increase H_2_O_2_ production and heme peroxidase activity [438,439].

The exact mechanisms involved in heme accumulation in the brain are still unclear. However, analysis of brain tissue from AD patients suggests that blood-derived hemoglobin may be the source of heme. Blood vessel destruction due to Aβ aggregation around brain vasculature may lead to localized brain hemorrhage and heme leakage [438,439,440,441].

Heme-related proteins/enzymes have been found to be altered in AD. Diverse studies have discovered the upregulation of hemopexin and heme oxygenases in AD [439]. Moreover, HO-1 upregulation and post-transcriptional modifications were detected in the brains of patients with AD in a previous study. Mouse studies demonstrated that HO-1 overexpression induces tau phosphorylation and facilitates aggregation [436,442,443]. HO-1 upregulation can induce Fe^2+^ excess in the cytoplasm, promote oxidative stress, and promote ferroptosis. Interestingly, APP was reported to bind and inhibit HO-1 and HO-2, where APP carrying mutations linked to FAD had a higher binding affinity and provided higher heme oxygenase inhibition [6]. Elevated HEBP1, a mitochondrial-associated heme-binding protein 1, is associated with AGE/RAGE-related neuronal damage, leading to neuronal loss in early AD. HEBP1-mediated apoptosis contributes to neuronal loss and dysfunction. Additionally, its cleavage product F2L regulates inflammation [438]. Moreover, the expression of BVR-A, an enzyme involved in the reduction of biliverdin to the antioxidant bilirubin during heme degradation, and its oxidative and nitrosative modifications, were elevated in samples from the hippocampus of patients with AD in a previous study. On the other hand, levels of phosphorylated BVR-A were decreased and associated with reduced cellular antioxidant activity in the cerebellum in AD [436]. Reduced BVR-A levels or impaired BVR-A activation contribute to the development of brain insulin resistance and metabolic alterations in AD. During AD progression, BVR-A Tyr phosphorylation is reduced, and BVR-A nitration is increased due to the elevated oxidative stress, a process that impairs the activation of the insulin signaling pathway and promotes CK1-mediated BACE1 recycling at the plasma membrane, where BACE1 cleaves APP and leads to elevated Aβ production. On the other hand, increased Aβ peptides trigger increased oxidative and nitrosative stress, leading to BVR-A impairment and resulting in a vicious cycle [444]. Moreover, studies have demonstrated that expression levels of the rate-limiting heme synthetic enzyme ALAS1 and the heme degradation enzyme HO-2 were selectively decreased in the hippocampi of AD brains. HO-2 and heme degradation were elevated in fully differentiated neuronal cells to support neuronal functions, but they were reduced after Aβ exposure. Furthermore, decreased heme metabolism, especially heme degradation, seems to be an early event in AD [445]. Therefore, targeting the HO-1 metabolites such as CO, bilirubin, and iron-mediated ferritin expression may suppress oxidative stress and ferroptosis, and mitigate AD progress [436].

Ferroptosis, an iron-dependent nonapoptotic form of regulated cell death, is potentially associated with AD [6,412]. The exact mechanism for iron-dependent ferroptosis is controversial, but it is believed that cytoplasmic Fe^2+^ reacts with membrane lipids and triggers a lethal lipid chain reaction and pore formation [6]. Iron imbalance may elevate hydroxyl free radicals and trigger neuroinflammation, leading to lipid peroxidation and redox dyshomeostasis, which are linked to the ferroptotic pathway [412]. Therefore, controlling iron levels and suppressing ferroptosis can be explored as a potential therapeutic approach for AD. Studies have demonstrated that ferroptosis can be suppressed by Nrf2-dependent induction of antioxidant enzymes such as GPx4, SOD2, and HO-1 [436]. Thus, exploring their therapeutic benefits may provide new insights into the treatment of AD pathology. Additionally, diverse studies have reported that using different iron chelators can rescue memory deficits, Aβ and tau pathologies, and neurodegeneration in AD patients [6]. Thus, targeting iron and heme might be a therapeutic strategy to overcome AD.

### 5.2. Calcium and Copper Imbalance

Dyshomeostasis of other minerals has been implicated in AD development. An analysis of the plasma of AD patients reported an increased phosphorus percentage but decreased levels of calcium, iron, copper, zinc, and selenium. Elevated serum phosphate is associated with increased pro-inflammatory cytokines and may contribute to escalating inflammation in AD [426].

Serum calcium levels are regulated by APOE, and the increase in neuronal calcium influx seems to correlate with the elevation of Aβ [446,447]. Dysregulation of calcium homeostasis may be the basis for APOE4 neurotoxicity in AD [426]. Mitochondrial Ca2+ overload in AD seems to depend on amyloid plaque deposition [133,448]. On the other hand, RNA-seq and microarray analyses showed an alteration of many genes involved in mitochondrial Ca2+ transport in AD brains. Several genes involved in mitochondrial Ca2+ influx were downregulated, while many of those involved in Ca2+ efflux were upregulated [448]. Even though this may result in decreased mitochondrial Ca2+ levels, it is believed to be a compensatory mechanism to avoid mitochondrial Ca2+ overload in the AD brain [449].

Copper (Cu) is a redox-active metal implicated in diverse brain metabolic activities [450,451,452]. Mutation of the Cu transporter *ATP7B* (*K832R*) increases the risk of AD and causes loss of its function [6,453,454]. Reduced plasma copper and ceruloplasmin, a cuproenzyme, levels correspond with impaired cognitive performance in AD patients [426]. Other studies also found low levels of SOD1, another cuproenzyme [6,455]. Similarly, APP overexpression in an AD mouse model induced copper deficiency, which is associated with impaired superoxide dismutase and ceruloplasmin preceding amyloid neuropathology appearance [426]. Other studies demonstrated that Cu concentrates with other metals in amyloid plaques. Ionic copper and iron facilitate Aβ aggregation and are involved in ROS generation [456,457,458,459]. Cu distribution in the AD brain is diverse; extracellular Cu^2+^ pooling can promote Aβ precipitation under acidic conditions and it can become trapped in the extracellular plaques, whereas intracellular copper deficiency may promote Aβ production [6]. Copper also upregulates APP expression [460,461]. Cell culture studies involving copper depletion reported increased Aβ generation [462,463], whereas dietary supplementation or upregulation of intraneuronal copper in APP transgenic mice suppressed Aβ production [455,464]. Copper–Aβ interaction can recruit substrates such as cholesterol to produce hydrogen peroxidase and promote oxidative stress, causing neurotoxicity [465,466,467,468,469]. Presenilin was reported to influence cellular copper turnover [470]. Moreover, in vitro and in vivo studies showed copper can also bind to tau protein, promote its aggregation, and induce the generation of hydrogen peroxidase, as observed in the copper ion–neurofibrillary tangle complexes within the neurons in AD brains [471,472,473]. Furthermore, copper modulates cyclin-dependent kinase 5 tau phosphorylation [6].

### 5.3. Zinc Imbalance

Zinc (Zn) is an essential element for brain function, concentrated in the gray matter and involved in neurotransmission [463,474,475]. Ionic zinc has been shown to induce histological amyloid structures. Zn^2+^ binds to Aβ, inducing its aggregation and deposition. The Aβ-zinc complex is resistant to proteolysis and promotes the stability of Aβ aggregates [476,477,478]. The slow turnover of synaptic Zn^2+^ released during glutamatergic synaptic transmission may induce amyloid deposition [6]. Furthermore, decreased levels of the zinc transporter ZnT3 in cortical tissues, combined with the Zn^2+^ trapped by extracellular Aβ aggregates, may cause a deficiency of intracellular Zn^2+^ and impair neuronal function [479,480]. Moreover, zinc may inhibit β-secretase cleavage activity of APP, and increased zinc may limit Aβ production, but it may also promote Aβ_43_ production through a different mechanism [6,463,481]. Zinc may also increase PSEN1 expression and affect APOE stability, particularly APOE4 [6,482]. Moreover, Zn^2+^ promotes tau phosphorylation and aggregation [483,484,485]. Despite the growing evidence that zinc dyshomeostasis inhibits neuronal function, its connection to neurodegeneration in AD is still unclear and requires extensive research [6].

### 5.4. Selenium Imbalance

In a previous study, selenium (Se) treatment of neuronal cells showed decreased Aβ production by reducing the BACE1 activity, protected cells against Aβ toxicity, and reduced 4-hydroxynonenal, a marker of ferroptosis [486]. Furthermore, Se plays an important role in ferroptosis and has been implicated in AD pathology [6]. Se is part of selenocysteine, which forms part of the active site of glutathione peroxidase 4 (GPx4), which is involved in ferroptosis [487,488]. Low Se levels have been reported in the serum and temporal cortex of AD brains [6,489,490,491,492]. Low Se levels in plasma were associated with an elevated risk of cognitive decline [493]. Additionally, selenium levels in the brain were found to be lower in APOE4 carriers [491]. Nutritional and environmental factors may modulate mineral levels in the serum and disease stage. Therefore, further research is required to understand the relationship between metal plasma levels and AD pathology.

## 6. Mitochondrial Dysfunction and Oxidative Stress in AD

An imbalance between the production of ROS and antioxidant defenses is known as oxidative stress [494]. There is evidence of significant levels of oxidative stress in AD [495,496]. A rise in oxidative stress is also reported as part of the aging process, but in the AD brain, oxidative stress is considerably higher compared to elderly controls of the same age. Mitochondria are the main source of oxidative stress. It has been proposed that defective mitochondria are less capable of generating adenosine triphosphate (ATP) but more capable of producing ROS, which may be a significant contributor to the oxidative imbalance observed in AD [21,497,498]. Increased ROS levels promote the transcription of proinflammatory genes and the production of cytokines such as IL-1, IL-6, and TNF-α, as well as chemokines, which result in neuroinflammation. On the other hand, inflammatory responses activate microglia and astrocytes to produce high levels of ROS. Therefore, neuroinflammation can be seen as both a cause and a result of persistent oxidative stress. The accumulation of Aβ peptide in the brain leads to the development of NFTs, inflammatory responses, increased oxidative stress, and mitochondrial dysfunction, which are the root causes of cell death and dementia [17,499]. Mitochondrial dysfunction is among the most noticeable and earliest indications of AD. It has been found that almost every component of mitochondrial activity is altered in AD [7]. Consistent aspects of mitochondrial dysfunction in AD include damaged mitochondrial bioenergetics, increased oxidative stress, and a disturbed mitochondrial genome. The significance of these defects in initiating mitochondrial dysfunction may vary depending on the specific biological, environmental, and genetic features of each patient with AD. However, any one of these abnormalities can lead to the other two, exacerbating neuronal dysfunction and neurodegeneration [47].

ATP, which is produced predominantly by mitochondria, is the cell’s primary source of biological energy production [500,501]. Decreased ATP generation and the formation of ROS are all consequences of mitochondrial energy metabolism dysfunction [502,503]. Three mitochondrial enzyme complexes have been found to have low activity in AD: cytochrome oxidase (COX), the pyruvate dehydrogenase complex (PDHC), and *α*-ketoglutarate dehydrogenase complex (KGDHC) [498,504,505,506]. Genetic abnormalities may cause mitochondrial enzyme abnormalities in the AD brain, and other AD-related processes, including oxidative stress, may affect mitochondrial enzymes. COX, PDHC, and KGDHC may be crucial steps in the chain of events that results in AD, regardless of whether or not there are primary genetic problems or subsequent events induced by oxidative stress [498].

### 6.1. Protein Oxidation, Lipid Oxidation, DNA Oxidation, and Glycoxidation

During the progression of AD, brain tissue in patients is subject to oxidative stress, including protein oxidation, lipid oxidation, DNA oxidation, and glycoxidation [507,508,509,510]. Protein carbonyl production is a significant marker of protein oxidation, which can be brought on by direct free radical attack on certain amino acid side chains, from the products of glycation and glycoxidation, or from products of lipid peroxidation interacting with proteins. The elevated levels of protein carbonyl and 3-NT in the AD brain, along with studies demonstrating that antioxidants can inhibit Aβ (1-42)-induced protein oxidation in neurons, are consistent with the idea that Aβ-induced protein oxidation may contribute to some of the neurodegeneration in the AD brain [511,512,513,514,515,516]. There is strong evidence that lipid peroxidation plays a key role in the neurodegeneration of AD. Lipid peroxidation caused by free radicals is common in AD. Free radical antioxidants prohibit lipid peroxidation in the membranes of brain cells, which is caused by Aβ. Taken together, lipid peroxidation caused by Aβ may contribute to the neurodegeneration of the AD brain [517,518,519]. In a previous study, nuclear DNA (nDNA) contained levels of oxidized bases about 10-fold lower than those found in mitochondrial DNA (mtDNA) [520]. The neurodegeneration found in AD may be influenced by the considerable oxidative damage to nDNA and mtDNA in the disease [521]. According to research on DNA oxidation, 8-Hydroxy-2-deoxyguanosine (8-OHdG) plasma levels are elevated in AD patients, indicating that this oxidized product could be employed as an AD diagnosis marker [522,523]. Accelerated oxidation of glycated proteins (glycoxidation) results in the buildup of extracellular advanced glycation end products (AGEs) in AD [507,524]. In AD brains, AGE levels are increased, and AGEs can promote the production of Aβ. AGEs may cause tau hyperphosphorylation at a number of AD-related locations as well as problems in spatial memory [525,526].

### 6.2. Aβ, tau, OXPHOS, and Mitochondrial Dysfunction

APP, through the creation of Aβ, alters the balance of mitochondrial fission and fusion, leading mitochondria to fragment and distribute in an abnormal way, which effects mitochondrial and neural dysfunction [51,527,528]. Neurons serve as the most fundamental units in the nervous system to carry out the exchange of information. They have the capacity to receive, integrate, transmit, and export information. In order to maintain appropriate neuronal activity, mitochondria, the organelles responsible for giving energy to neurons, are carried to the dendrites and axons that require the most energy through the microtubule structure [529]. The balance of mitochondrial fission and fusion is indeed altered in AD neurons, which is thought to contribute to mitochondrial dysfunction and neurodegeneration in vivo [530]. Mitochondrial dysfunction caused by the formation of ROS increases Aβ production, creating a vicious cycle between mitochondrial dysfunction and ROS, as well as the harmful effects of Aβ [531,532]. The Aβ peptide needs functioning mitochondria to cause cell toxicity, as mitochondria are both the source and the target of ROS-induced neuronal damage, according to Cardoso et al. [533]. Mitochondrial dysfunction is the primary contributor to Aβ deposition, synaptic degeneration, and NFT generation in sporadic late-onset AD. The main distinction between sporadic and familial AD is that in the latter, Aβ appears to be the primary pathogenic event that results in subsequent mitochondrial malfunction [534,535].

A potential therapeutic target for mitochondria has been investigated [536,537]. Insulin has been found to stop the reduction in mitochondrial oxidative phosphorylation efficiency and prevent a rise in oxidative stress in the presence of the Aβ peptide, enhancing or maintaining the function of neurons in AD [536]. Additionally, treatment with coenzyme Q10 (CoQ10) can prevent brain mitochondrial dysfunction caused by Aβ_1–40_, hence preventing the severe energy deficit underlying AD pathogenesis [537]. Moreover, evidence indicates that insulin regulates the metabolism of Aβ and tau proteins [538,539]. Research has established that hyperphosphorylated tau impairs mitochondrial transport, resulting in an energy deficit and oxidative stress at the synapses, resulting in neurodegeneration [51]. Tau accumulation and phosphorylation are consequences of mitochondrial malfunction. In the meantime, pathogenic tau disrupts mitochondrial dynamics and function as well as axonal transport. This creates a vicious cycle of increasing mitochondrial malfunction and tau pathology, influencing neuronal and synaptic function, and resulting in memory loss and cognitive issues in AD. Defective mitochondrial function markers include reduced ATP synthesis, enhanced ROS generation, defective OXPHOS complexes, and antioxidant enzymes [51,540,541,542]. While simultaneously monitoring cellular health to make a prompt choice (if required) to begin cell death, mitochondria are crucial for producing the energy that powers regular cellular function. Mitochondria are crucial for neuronal function since neuronal cells have a limited scope for glycolysis and rely heavily on aerobic OXPHOS for their energy needs. However, OXPHOS is the primary source of harmful free radicals, which are byproducts of regular cellular respiration [497,543,544]. Oxidative stress may facilitate tau polymerization and phosphorylation while increasing the construction and accumulation of Aβ. This sets up a vicious loop that encourages the development and spread of AD [17]. Figure 4 provides a summary of the effects of mitochondrial dysfunction, oxidative stress, tau, and Aβ in AD.

## 7. Altered Signaling Pathways in the AD Brain

A pathway found to be altered in AD is the cyclophilin A (CypA)-NF-κB–matrix-metalloproteinase-9 (MMP-9) pathway. Activation of the CypA-NFκB-MMP-9 pathway in pericytes of APOE4 mouse models may cause BBB tight junction and basement membrane protein degradation, inducing BBB disruption [117,545,546,547]. Additionally, diverse studies indicate that dysfunctional brain insulin signaling, and subsequent glucose metabolism disruption, are associated with sporadic AD [116,548]. Glucose uptake and utilization are reduced in the hippocampus, parietotemporal cortex, and/or posterior cingulate cortex in AD subjects [116,117,549]. Decreased levels of glucose transporter 1 (GLUT1) in cerebral microvessels have been found in AD brains. Glut1 transports glucose across the BBB from the blood to the brain and maintains brain capillary networks, cerebral blood flow (CBF), and BBB integrity [117].

Brain glucose dyshomeostasis may occur several years before the beginning of symptoms. Increased plasma fasting glucose is associated with elevated concentrations of brain tissue glucose, which is linked with severe Aβ plaque deposition and neurofibrillary pathology in brain areas susceptible to AD pathology [550]. However, rate-controlling glycolysis enzymes (hexokinase, phosphofructokinase, and pyruvate kinase) activity was found to be reduced in the inferior temporal gyrus and the middle frontal gyrus of AD patients, and was associated with acute neurofibrillary and amyloid pathology. All these observations suggest that regionally specific abnormalities in glycolytic flux may result in brain glucose accumulation and AD development [116,550]. Alterations in brain glucose concentration in AD may be explained by the low protein levels of GLUT3, a neuronal glucose transporter, which are associated with increased neuritic plaque and neurofibrillary tangle pathology, possibly during the early stages of AD development [550,551,552].

Insulin resistance plays an important role in promoting cognitive dysfunction, and affects AD pathogenesis [553,554,555,556]. Postmortem brains show increased serine phosphorylation of IRS-1 and JNK expression with Aβ colocalization. Aβ oligomers and a high-fat diet stimulate the JNK phosphorylation in the hippocampal neurons, leading to insulin signaling downregulation [557,558]. Despite the risk factor of insulin resistance for AD development, it may also aggravate AD pathology. Insulin affects Aβ metabolism, and Aβ interferes with insulin binding to its receptor. Additionally, peripheral insulin resistance was found to be associated with low hippocampal glucose metabolism and low gray matter volume. Moreover, insulin resistance may be aggravated in AD due to chronic inflammation [116,559,560,561].

Studies have associated NLRP3 inflammasome activation with NFT formation and Aβ deposits in AD pathology [15,562]. NLRP3 activation induces IL-1β production, promotes microglial synthesis, and releases proinflammatory cytokines and neurotoxic factors. However, an enhanced NLRP3 signaling pathway may induce elevated tau hyperphosphorylation, neurofibrillary tangle formation, synaptic dysfunction, and neuroinflammation [16,116,562]. Animal studies showed that the administration of an NLRP3 inhibitor improved cognitive impairment and hyperactive behavior by reducing Aβ deposition via increasing the microglia’s ability to clear Aβ deposits [16,563]. Therefore, the association of NLRP3 and AD pathology can be explained as an endless loop [116].

Another pathway compromised in AD pathology is the glutathione (GSH) pathway, which promotes lipid peroxidation, which induces ferroptosis. GSH and glutathione peroxidase (GPx) expression are downregulated in the frontal cortex and hippocampus and correlate with the impairment’s severity [6,564,565,566].

Another pathway found to be altered in AD is the ceramide pathway. Ceramides act as bioactive lipids, important regulators of synaptic function, and can affect the production of Aβ and tau phosphorylation. Elevated levels of ceramides were discovered in AD brains, suggesting the early implication of lipid metabolism in AD. Increased ceramide levels stimulate ROS generation, leading to neuronal death [567,568,569]. Enhanced levels of ceramides increase the levels of Aβ by stabilizing β-secretase, while Aβ induces an increase in ceramide levels by activating sphingomyelinases, which catabolize sphingomyelin into ceramide [567]. Moreover, studies found increased sphingomyelin levels in the CSF of prodromal AD patients [570,571,572,573]. An increased sphingomyelin/ceramides ratio can be a potential blood-based marker of AD progression. Elevated levels of ceramide are linked to JNK activation, which mediates cell death in AD. Ceramides decrease the phosphorylation of ERKs and MEKs, trigger JNK and p38 MAPK cascades, and induce neuronal death through the activation of caspase-3 and upregulation of c-jun, c-fos, and p53 [567,574,575,576].

Another pathway altered in AD is the Akt pathway. PI3k-Akt activation can protect against neuronal death in AD [577,578]. In a previous study, increased levels of active phosphorylated Akt were discovered in particulate fractions, with decreased Akt levels in cytosolic fractions of AD temporal cortex neurons, leading to increased activation of Akt in AD progression. Moreover, decreased levels of the Akt target p27^kip1^ and elevated phosphorylation levels of Akt substrates (GSK3β^Ser9^, tau^Ser214^, mTOR^Ser244B^) were detected. Furthermore, a loss and altered distribution of the major Akt- negative regulator PTEN were observed in AD neurons [579]. S-nitrosylation may inactivate PTEN in AD and explain the loss of PTEN. PTEN regulates neuronal development and survival, axonal regeneration, and synaptic plasticity in the CNS [580,581,582,583]. Neurons containing phosphorylated Akt and PTEN were depleted in hippocampal CA1 at the end stages of AD [579]. Moreover, PTEN accumulates in intracellular fibrillary tangles and co-localizes with tau and senile plaques [583].

## 8. Mathematical Models of AD Progression

Many mathematical models of the pathophysiology of AD have been developed during the past few decades. The majority of studies concentrate on the aggregation of the Aβ. Numerous models have been presented to investigate the spread of prion proteins [2,584,585,586,587]. For proper operation, a protein must appropriately fold into a suitable three-dimensional (3D) shape. Aggregates generated by misfolded proteins can occasionally form, and are extremely toxic to normal healthy cells. A clump of particular proteins has been associated with a variety of illnesses, including AD. The two proteins that combine to create protein aggregates in the brains of AD patients are Aβ and tau. According to Heller, these Aβ and tau aggregates hinder common brain activities [588]. Synthetic Aβ oligomers attach to cellular prion protein (PrPC). The spatial learning and memory of AD transgenic mice with intact PrPC expression are impaired. It has been claimed that PrPC-deficient AD transgenic mice accumulate Aβ but, nevertheless, exhibit normal survival and no decline in spatial learning and memory [589].

### 8.1. Deterministic Mathematical Models

Mathematical models of Aβ aggregation have been previously developed. There are some deterministic mathematical approaches to studying AD [2,585,586,587]. An elaborate kinetic model has been created that quantitatively explains the kinetics of Aβ self-association from the unfolded state. Nonlinear regression was used to fit the model to the experimental data and identify the parameters. The model incorporated data on changes in both the distribution of mass and the length of the protein aggregates. It took into account the co-existence of different species of the protein, including monomers, dimers, and larger aggregates. The model also provided mechanisms for the generation and elongation of the fibrils, while explicitly accounting for the filaments and fibrils themselves. As a result, the model was able to accurately represent the key features of the experimental data and provide a detailed and comprehensive quantitative description of the kinetics of the protein aggregates [586]. A compartmental mathematical model has been created to track the accumulation of Aβ protein in the brain, CSF, and plasma during AD. The analysis showed that the total Aβ burden in the brain is influenced by a dimensionless quantity known as the polymerization ratio, which is equal to the product of production and elongation rates divided by the product of fragmentation and loss rates. The finding suggests that the Aβ levels in all three compartments are likely going to decrease when production inhibitors are used. The presentation of straightforward formulae and numerical findings offers some insights into system behavior, which may be utilized to determine important transport, production, and clearance parameters as new clinical data become available [587]. Helal et al. [2] presented a mathematical model for the development of AD that utilizes differential equations to explain the dynamic production of Aβ plaques depending on concentrations of Aβ oligomers, PrPC proteins, and the Aβ-x-PrPC complex that is believed to create synaptic toxicity. The model of AD evolution is based on the idea that Aβ oligomers can exist bound and unbound to PrPC proteins. Dayeh et al. [585] presented a model that described a strategy to lessen the formation of oligomers, and created a discrete mathematical model for the accumulation of monomers into oligomers. This hypothesis is based on the idea that soluble oligomers are what cause AD. Following that, a stability analysis establishes the aggregation condition. For the purpose of creating oligomers, a formula for the necessary number of monomers is given.

### 8.2. Stochastic Mathematical Models and Potential Development of the Models for T Cells in AD

The deterministic models did not take into account the involvement of random factors. The neurobiological environment and the transfer of its signals are more random than deterministic, incorporating a variety of random elements. For instance, in the same environment, various cells and infectious virus reactions can produce various outcomes. Hu et al. [590,591,592] studied a stochastic AD model that took the effects of prions into account for the first time. They investigated the peculiar ergodic stationary distribution of the model to understand how AD progresses, and discovered that the system has a unique ergodic stationary distribution under weak conditions. This suggests that AD will likely continue after a person develops it. Later on, Li et al. [584] presented an AD model that takes into account two forms of Aβ, the interaction of oligomers with Prpc, and the use of anti-Aβ medications. The existence, singularity, and non-negativity of the solutions were examined. Additionally, it has been demonstrated that the model has a single globally asymptotically stable equilibrium, indicating that the medicine cannot totally cure AD. Finally, some numerical simulations have been provided, and the relative contributions to AD of the two forms of Aβ, Aβ_40_ and Aβ_42_, have been explored.

None of the earlier mathematical models took the development of AD in connection to the role of T cells into consideration. However, there are certain mathematical models that could be constructed for AD as an autoimmune illness. These studies used a set of ordinary differential equations (ODE) to describe CD4^+^ helper T cells’ immune responses in the existence of Tregs. In addition to formulas for the concentration of T cells, the concentration of Tregs, and antigenic stimulation of T cells, the equilibria of the corresponding ODE systems were also determined. This model, which uses two T cell clonotypes, was used to investigate how bystander T cell proliferation contributes to the development of autoimmune responses and the prevention of autoimmune disease. Hysteresis, which was apparent for the parameter values, also suggests that severe immune suppression may be needed for the therapy of autoimmunity. If this is the case, the system may not enter the managed steady state’s basin of attraction as a result of this treatment [354,593,594,595].

## 9. Conclusions

Millions of people worldwide suffer from AD, a neurological condition for which there are no effective treatments available. While estimates vary, experts believe that over 6 million Americans, the majority of whom are aged 65 or older, may have dementia brought on by AD. Researchers are still working to understand the complex brain alterations that occur in AD, which can begin years or even decades before symptoms appear. While there is no cure for AD at present, a deeper understanding of these changes may lead to the development of effective treatments. Common modifications observed in AD include Aβ aggregation, NFTs, BBB alteration, neuroinflammation, impaired metal ion homeostasis, mitochondrial dysfunction, oxidative stress, disruption of normal glial function, synaptic and neuronal loss, metal ion imbalance, genetic risk factors, cholinergic neuronal degeneration, and the immune system. These modifications, along with mathematical models of AD, shed important light on diverse strategies that can be used to combat AD pathology.

## Figures and Tables

**Figure 1 ijms-24-07258-f001:**
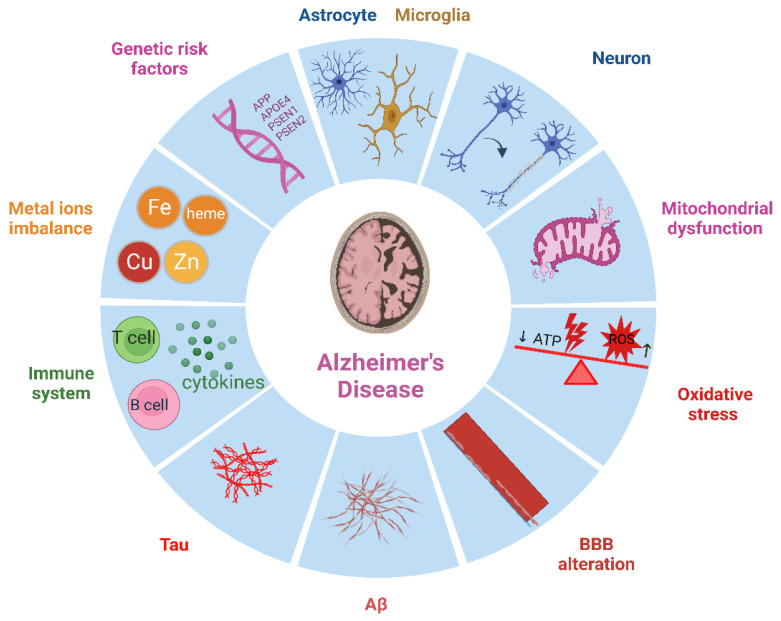
Multiple factors are associated with AD, including defective BBB, mitochondrial dysfunction, oxidative stress, metal ion imbalance, genetic risk factors, and the immune system. Additionally, AD pathogenesis involves T cells, Aβ, tau, microglia, and astrocytes. A comprehensive understanding of these factors and their interplay is crucial for advancing our understanding of AD and developing effective treatments. Created with BioRender (BioRender.com).

**Figure 2 ijms-24-07258-f002:**
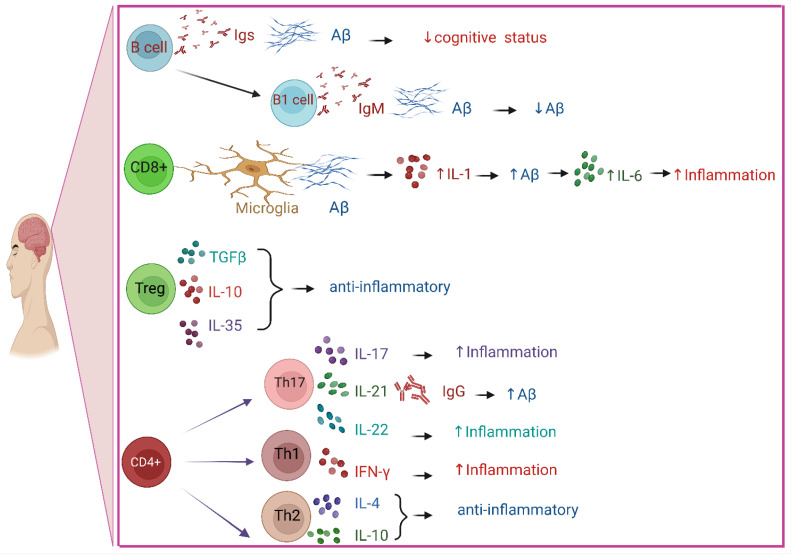
CD4^+^ T cells differentiate and proliferate into Th1, Th2, and Th17 cells. Th17 and Th1 cells secrete proinflammatory cytokines, but Th2 secrete ant-inflammatory cytokines. Depletion of Tregs worsens mice’s cognitive impairment. Immunosuppressive anti-inflammatory cytokines are released by Tregs. While it is thought that immunoglobulin encourages and accelerates the uptake of Aβ plaques by microglia, it is also thought that the Aβ–IgG complex in the CSF has a negative impact on the cognitive health of AD patients. Created with BioRender (BioRender.com).

**Figure 3 ijms-24-07258-f003:**
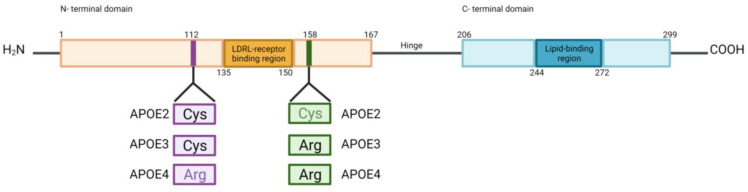
Amino acid residue substitutions of APOE isoforms. APOE consists of an N-terminal domain that contains an LDLR receptor-binding region connected by a hinge region with a C-terminal domain that contains a lipid-binding region. Positions 112 and 158 show amino acid substitutions among APOE isoforms. Cysteine (Cys). Arginine (Arg). Created with BioRender (BioRender.com).

**Figure 4 ijms-24-07258-f004:**
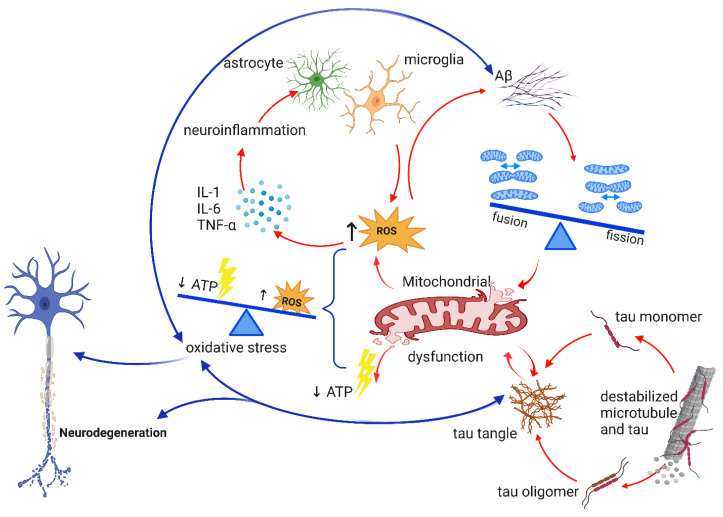
Production of Aβ alters the equilibrium between mitochondrial fusion and fission, leading to mitochondrial malfunction. Through the generation of ROS, mitochondrial malfunction causes an increase in Aβ production, starting a vicious cycle. Increased ROS levels increase the production of cytokines, including TNF-α, IL-1, and IL-6, which leads to neuroinflammation. On the other hand, inflammatory responses cause microglia and astrocytes to become highly active and generate high levels of ROS. Tau pathology and growing mitochondrial dysfunction are a vicious loop that affects synaptic and neuronal function. Created with BioRender (BioRender.com).

**Table 1 ijms-24-07258-t001:** A summary of the effects of tau and Aβ, two main players that are associated with the development of AD.

Biomarkers	Effects	References
**Aβ**	results in heme deficiency, which changes: •Homeostasis of zinc, iron, and Ca^2+;^ •APP; •NOS. causes oxidative stress and inflammation and vice versa produces hyperphosphorylated tau develops mitochondrial dysfunction and vice versa results in neurodegeneration	Atamna et al. 2002, [43] Chai et al. 2020, [26] Gotz et al. 2004, [45]; Ittner et al. 2011, [46] Morley et al. 2014, [37] Wang et al. 2020, [47]
**Tau**	NFTs build up instability of microtubule causes oxidative stress and vice versa results in neurodegeneration buildup of tau tangle results in: •Neuroinflammation; •impaired synaptic function; •dysfunctional autophagy; •dysfunctional mitochondria.	Miao et al. 2019, [48] Brion 1998, [49] Rawat et al. 2022, [50] Eckert et al. 2011, [51] Rawat et al. 2022, [50]

## Data Availability

No new data were created or analyzed in this study. Data sharing is not applicable to this article.

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
