# Peer review of "Recent Development in the Understanding of Molecular and Cellular Mechanisms Underlying the Etiopathogenesis of Alzheimer’s Disease"

_ijms, 2023, doi:10.3390/ijms24087258_

Round 1

Reviewer 1 Report

Reviewer comments and suggestions

The author of this study discusses the most common alterations associated with AD and promising therapeutic targets for AD drug discovery. They also highlight the role of heme in AD development and summarizes mathematical models of AD, and mathematical models of the effect of Aβ on AD. The study also included potential treatment strategies that these models can offer in clinical trials.

Below are the comments for this paper to be incorporated in the revised version of the manuscript. 

  1. The title suggests AD drug discovery therapeutics but I could not find this information in the manuscript, so it would be better to modify the title based on the content
  2. The first time AD was used in the introduction should be in full form.
  3. please mention a few references “According to various studies, oxidative stress may increase the generation and aggregation of Aβ and ease the polymerization and phosphorylation of tau”. 
  4. how it happens please explain. Tau oligomers also travel from one neuron to another [25].
  5. I noticed that special characters used in the text of the manuscript were typo-errored, please proofread your manuscript once again. Such as TNF-α, Aβ plaques
  6. Needed references. “ Several studies demonstrate a link between diet, obesity, and AD. Obesity increases the risk of cognitive decline and AD development by six-fold in adulthood and midlife.”
  7. what does it mean? “Discrepancies among samples from different populations suggest further studies are required to elucidate its role in AD pathology [255]”
  8. Section 8.1 briefly describes mathematical modeling

Reviewer 2 Report

Manuscript Number: ijms-2228405

Title: Illuminating cellular and molecular mechanisms for AD drug discovery

By A. Afsar et al.

The work has some advantages, but from my point of view, it requires corrections before publishing.

Here are some suggestions:

1.      In the Introduction part, the significance, purpose, and innovation of this review cannot be found.

It has already been other reviews on related topics. For example:

Guo, T.; Zhang, D.; Zeng, Y.; Huang, T.Y.; Xu, H.; Zhao, Y. Molecular and cellular mechanisms underlying the pathogenesis of Alzheimer’s disease (Review). Mol Neurodegener 2020, 15, 40, doi:10.1186/s13024-020-00391-7.

Chen, X.-Q.; William C. Mobley, W.C. Alzheimer Disease Pathogenesis: Insights From Molecular and Cellular Biology Studies of Oligomeric Aβ and Tau Species (Review). Front Neurosci 2019, 13, doi:10.3389/fnins.2019.00659

Please, add these references in the Introduction and a corresponding comment.

2.      In the Introduction part, it is necessary to add that the genetic risk factors associated with AD were presented in Section 4, while the study of imbalances of metal ions such as heme, Fe, Cu, and Zn was given in Section 5.

3.      Please, provide a list of abbreviations.

Reviewer 3 Report

Alzheimer’s disease (AD) causes 60-70% of all cases of dementia. Incidence is expected to increase in the coming decades mostly due to the aging world population and the lack of effective treatments. In this review, authors discuss the most common biological and biochemical alterations that research has found to be associated to the disease.

Authors provide a comprehensive review that covers the most critical aspects of AD. The manuscript should be of the interest of IJMS readers upon minor revisions.

11)     English is generally fine, but a minor revision should be in order. Some verb tenses sound inappropriate, and some words are repeated with a short separation.

22)     Line #1 of the introduction should state the entire name of the disease “Alzheimer’s disease” before the abbreviation AD.

33)     Figure 1 caption repeats “genetic risk factors” twice in the second line.

44)     Is the description of T cells and microglia interaction relevant in Figure 1 caption?

55)     Table 1 needs to have columns separated by a line. Also, bullet points should be aligned.

66)     Alpha and beta symbols need to be corrected throughout the manuscript, as they don’t display properly.

77)     In section 2.3 authors discuss mostly diet, which seems to be more related to section 2.2 than to section 2.3. Please revise.

88)     Why are words capitalized in section 2.5.3?

99)     Section 5.2 includes an entire paragraph about Ca2+. Authors should then consider rephrasing the headline of the section as “Copper and calcium imbalance,” or to separate the calcium discussion into a different section.

Reviewer 4 Report

The topic and the scientific information and presentation in the article are importance, however, the writing style (English) and some word choices make the article difficult to follow and to appreciate; some of which are pointed out below. 

1, In the 'Introduction, section, line 2, the word 'efficient' should be replaced by the word 'effective'.

(2). Also, in line 2, the word 'deepen' may be more appropriately changed to "increase".

(3). In line 9, the words "blood brain barrier" should be spelt out, according to the general rule of spelling out a 'long' word when it is first used.    

(4) In line 14, change the phrase: "BBB stability decreases, enhancing its leakage" to "BBB barrier stability decreases, causing its leakage." or 'causing it to leak'.

(5) The meaning of and the relevance of the sentence in line 17-19, in the ‘Introduction’ section are not clear; notably the phrase “shortage of Aβ overproduction”, since ‘shortage’ and ‘overproduction’ are somewhat contrasting situation.  

(6) On page 3, line 5, under the heading; '2. Neuropathological hallmarks of AD". Instead of writing, "Aβ is naturally created", it seems more appropriate to write, “Aβ is naturally produced”.   

These and several more writing-related errors, which mean that the manuscript should be carefully reviewed for errors in writing. It would be good to have the article pre-reviewed for 'English writing' before any subsequent re-submission. 

Page 5, line 10: what is ‘complex Iv-dependent respiration’?

Additionally:

A ’flat coil-like symbol’ occurs on several pages, notably pages 6, 7. 

Page 8, line 2 of paragraph 2 the sentence that read: “Acetylcholinesterase hydrolysis acetylcholine to choline and acetate” should be changed to read “Acetylcholinesterase hydrolyses acetylcholine to choline and acetate”.

The statement that, “The BBB acts as a barrier to keep neurons working at their best”, seems very elementary to use the phrase "at its best, without explanation. Moreover, the sentence that follows, and stated that “The BBB is a cellular barrier that separates the blood from the brain’ and protects the CNS from circulation by preventing the entry of neurotoxic substances”; seems to be limited in scope, since neurotoxic substances can reach the brain, in spite of the BBB.

On page 22. ‘Section’ 7. with the title "Altered signaling pathways in the AD brain (NGS-bioinformatics)". In the 2nd line that follow the words "BBB thigh junction” occurs and should be corrected to “BBB tight junction”.   

The Figures/tables.

Figure #1. It does not appear that figure #1 adds much information nor clarity to the article. Moreover, the words:  ‘genetic risk factors’ are presented twice, and the abbreviation, ‘APC’ should be written out as the word.

Table 1. The table does not look as though it is professionally prepared, since (1) the lines that contain the words are unevenly spaced from the margin, (2) the items Aβ and Tau should have a heading and (3), the title should add more explanation/relevance to the contents of the table, rather than merely “Main two players in AD”.  So, the table seems to present a state of disorganization.  

In summary, the information and scientific presentation in the article are importance, however, the writing style (English) and especially some word choices make the article difficult to follow and the content not easily appreciate.  
